# What is the impact of multimorbidity on the risk of hospitalisation in older adults? A systematic review study protocol

Luciana Pereira Rodrigues ,[1] Andréa Toledo de Oliveira Rezende,[1] Letícia de Almeida Nogueira e Moura,[1] Bruno Pereira Nunes,[2] Matias Noll,[3] Cesar de Oliveira,[4] Erika Aparecida Silveira[1,5]

[1]Postgraduate Program in Health Sciences, Universidade Federal de Goiás, Goiânia, Goiás, Brazil
[2]Department of Nursing in Collective Health and the Postgraduate Program in Nursing, Federal University of Pelotas, Pelotas, Brazil
[3]Instituto Federal Goiano, Ceres, Goiás, Brazil
[4]Department of Epidemiology and Public Health, University College London, London, UK
[5]Affiliate Academic, Department of Epidemiology and Public Health, University College London, London, UK

**Correspondence to**
Erika Aparecida Silveira;
erikasil@terra.com.br

## ABSTRACT

**Introduction** The development of multiple coexisting chronic diseases (multimorbidity) is increasing globally, along with the percentage of older adults affected by it. Multimorbidity is associated with the concomitant use of multiple medications, a greater possibility of adverse effects, and increased risk of hospitalisation. Therefore, this systematic review study protocol aims to analyse the impact of multimorbidity on the occurrence of hospitalisation in older adults and assess whether this impact changes according to factors such as sex, age, institutionalisation and socioeconomic status. This study will also review the average length of hospital stay and the occurrence of hospital readmission.

**Methods and analysis** A systematic review of the literature will be carried out using the PubMed, Embase and Scopus databases. The inclusion criteria will incorporate cross-sectional, cohort and case–control studies that analysed the association between multimorbidity (defined as the presence of ≥2 and/or ≥3 chronic conditions and complex multimorbidity) and hospitalisation (yes/no, days of hospitalisation and number of readmissions) in older adults (aged ≥60 years or >65 years). Effect measures will be quantified, including ORs, prevalence ratios, HRs and relative risk, along with their associated 95% CI. The overall aim of this study is to widen knowledge and to raise reflections about the association between multimorbidity and hospitalisation in older adults. Ultimately, its findings may contribute to improvements in public health policies resulting in cost reductions across healthcare systems.

**Ethics and dissemination** Ethical approval is not required. The results will be disseminated via submission for publication to a peer-reviewed journal when complete.

**PROSPERO registration number** CRD42021229328.

## Strengths and limitations of this study

► This review will not present restrictions in terms of language and year of publication regarding the articles that will be included in the search.
► The review process for selection and inclusion of the articles will be performed separately by two researchers and any disagreements will be mediated by a third reviewer in order to assure a consistent and strict application of the inclusion and exclusion criteria.
► Another strength is that there is no registry of previews systematic reviews on the association between multimorbidity and hospitalisation focused on older adults.
► Additionally, this study will include the three most commonly used definitions of multimorbidity in order to reach a greater number of studies related to the theme.
► The limitation of this review might be the difficulty to conduct a meta-analysis since the potential selected studies may have used different definitions for multimorbidity as well as subgroups analysis stratified by age and sex.

## INTRODUCTION

Chronic non-communicable diseases are considerably increasing globally, particularly in low-income and middle-income countries, due to lifestyle changes, accelerated urbanisation and increased longevity.[1] Multimorbidity is defined as the co-occurrence of two or more chronic conditions in the same individual.[2–6] Recent studies have also used the term 'complex multimorbidity', which is defined as 'the co-occurrence of three or more chronic conditions affecting three or more different body systems in one person, without defining an index chronic condition'.[7]

Despite the growing research in the field, there are still some difficulties in relation to the specific definitions of multimorbidity and comorbidity.[8–10] Comorbidity is defined as the co-occurrence of chronic conditions from an index disease.[11] Multimorbidity, however, focuses on the interaction of several coexisting diseases and not on a predominant condition.[12] Although the terms are similar, they should not be used interchangeably. In

the present study, we will use the terms multimorbidity, which was incorporated into Medical Subject Headings (MeSH) in 2018, and complex multimorbidity.[5 13]

The prevalence of multimorbidity has increased worldwide and affects over 50% of older adults, as increased life expectancy and longevity are associated with a greater disease burden.[14 15] In this age group, the simultaneous occurrence of two or more chronic diseases is more common than in younger individuals, leading to the associated use of multiple medications and increasing the likelihood of adverse effects, mortality and a higher number of hospitalisations.[3 16–18]

The use of healthcare services, including hospitalisation, is associated with specific factors such as sociodemographic variables, particularly socioeconomic status and the co-occurrence of chronic conditions.[19] In a study of sixteen European countries, multimorbidity was found to be associated with a greater number and length of hospitalisation episodes in individuals aged 50 years or older.[20] Another study conducted in southern Brazil found an association between multimorbidity and hospitalisations in individuals aged ≥60 years.[21] As a result, some studies have found an association between multimorbidity and hospitalisations,[20 21] whereas other studies have found an association only in participants over 80 years of age.[22 23] Therefore, there is evidence to indicate that greater hospitalisation rates are found in people with multimorbidity, however, the evidence from previous studies varied and they were not standardised.[24 25]

Multimorbidity negatively impacts on health and quality of life, and increases expenditure on healthcare.[26 27] It is, therefore, crucial to obtain a deeper understanding of the impact of multimorbidity on hospitalisations later in life. This will potentially guide the development of public health policies and improve the care and treatment of older adults with multimorbidity. It will also lead to reductions in the number of hospitalisation episodes and their associated negative consequences, such as loss of mobility,[28] stress, risk of falling, poor mental health,[29] cognitive impairment, social isolation and decreased quality of life.[30] Furthermore, older adults with multimorbidity use more health services, which is associated with an increased risk of death.[31]

Despite the importance of the problem, only one systematic review on this topic (published in 2011) was identified.[32] This systematic review focused on individuals with multiple chronic conditions and analysed several outcomes, including the use (number of medical consultations, use of medication and use of hospital services) and cost of the health services. However, this systematic review did not provide further details on the impact of multimorbidity on hospitalisations in older adults. The present study aims to provide a better understanding of this topic. More importantly, many articles presenting new evidence on the subject have been published in the last 10 years.

The impact of multimorbidity on hospitalisation episodes in older adults has not been previously investigated in a systematic review and meta-analysis. Therefore, the objective of this systematic review study protocol is to analyse the impact of multimorbidity on the occurrence of hospitalisation in older adults and assess whether this impact varies by sex, age, institutionalisation and socioeconomic status. This study will also review the average length of hospital stay and the occurrence of hospital readmission. The relevance of the topic for public healthcare and particularly geriatrics/gerontology relates to the increasing ageing population globally and its effect on society, particularly with regards to health.[33]

## METHODS AND ANALYSIS

This systematic review study will be conducted and reported following the Preferred Reporting Items for Systematic Reviews and Meta-Analyses approach.[34] The review will adopt the Population, Exposure, Comparator, and Outcome (PECO) structure recommended for systematic reviews.[35] The PECO structure was defined as 'P' (older adults in the community), 'E' (multimorbidity), 'C' (associated factors), and 'O' (hospitalisation). Any changes that occur during the study will be reported through the PROSPERO website and the final manuscript.

---

**Box 1  Search strategy for studies on multimorbidity and hospitalisation in older adults**

1 = 'multiple chronic conditions' OR multimorbidity OR multimorbidit* OR 'multi morbidit*' OR multi-morbidity OR 'chronic conditions multiple' OR 'multiple chronic health conditions' OR 'multiple chronic medical conditions' OR 'multiple chronic illnesses' OR 'chronic illnesses multiple' OR 'multiple chronic diseases' OR multidisease OR multidiseases OR 'multiple condition' OR 'complex needs' OR 'concurrent chronic conditions' OR 'concurrent chronic diseases' OR 'concurrent chronic disorders' OR 'concurrent chronic health conditions' OR 'concurrent chronic illnesses' OR 'concurrent chronic medical conditions' OR 'multiple chronic disorders' OR 'simultaneous chronic illnesses' OR 'simultaneous chronic medical conditions'

2=elderly OR elder OR aged OR ageing OR aging OR 'old adults' OR 'older adults' OR 'older people' OR 'old people' OR 'geriatric' OR 'aged patient' OR 'aged people' OR 'aged person' OR 'aged subject' OR 'elderly patient' OR 'elderly people' OR 'elderly person' OR 'elderly subject' OR 'senior citizen' OR senium

3=hospitalisation OR 'patient readmission' OR inpatients OR hospitalized OR 'health services' OR 'medical assistance' OR 'intensive care unity' OR 'health care utilization' OR 'length of stay' OR 'short stay hospitalization' OR' hospital admission' OR 'admission, hospital' OR 'patient admission' OR 'health care use' OR 'health care utilisation' OR 'health resource utilization' OR 'health service use' OR 'health service utilisation' OR 'health service utilization' OR 'health service utilization pattern' OR 'health services use' OR 'health services utilisation' OR 'health services utilization' OR 'utilization, health care' OR 'hospital patient' OR 'hospitalised patient' OR 'hospitalised patients' OR 'hospitalized patient' OR 'hospitalized patients' OR 'in-hospital patient' OR 'in-hospital patients' OR 'in-patient' OR 'in-patients' OR inpatient OR 'patient, hospital'

4=1 AND 2 AND 3

---

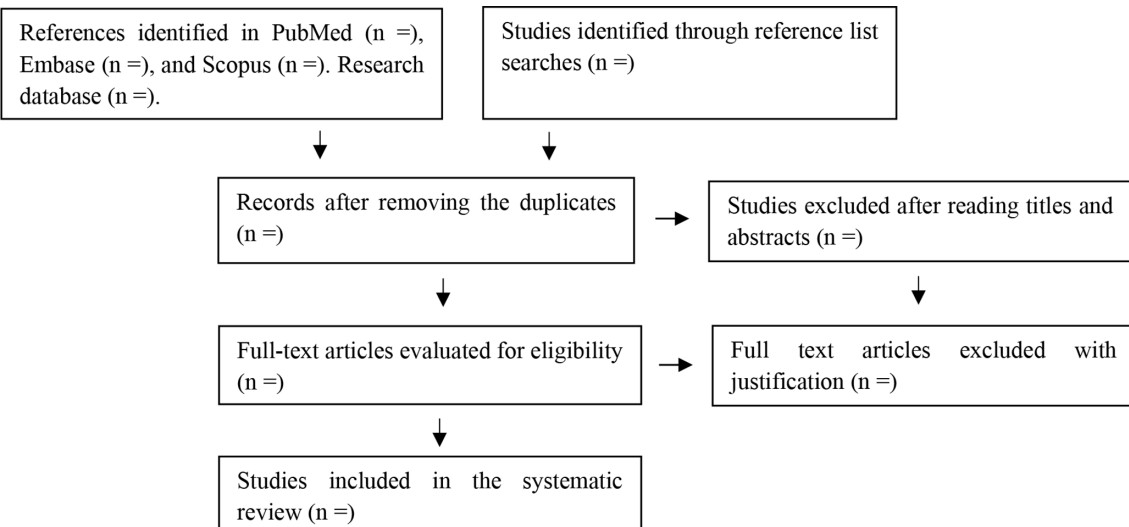

**Figure 1** Preferred Reporting Items for Systematic Reviews and Meta-Analyses flow chart: search and inclusion process of studies to include in systematic review of the multimorbidity impact on the risk of hospitalisation in older adults.

## Search strategy and eligibility criteria

From February 2021 to April 2021, searches will be carried out in the PubMed, Embase and Scopus databases by two independent researchers. There will be no restriction on the language and year of publication of the included studies and articles published until 30 April 2021 will be considered. In the search strategy, MeSH terms and relevant keywords related to multimorbidity, hospitalisation and older adults will be used, with the objective being to cover all articles on the topic (box 1).

The inclusion criteria are: (a) cross-sectional, cohort and case–control studies; (2) study subjects aged 60 years or older; (3) use of the definition of multimorbidity of ≥2 or ≥3 coexisting illnesses and complex multimorbidity and (4) hospitalisation as the outcome.

There are different ways to standardise and define multimorbidity. Despite the suggestion of Fortin *et al*[36] to apply the definition as being the co-occurrence of three or more chronic conditions in the same individual, to better identify individuals who need more healthcare (as is often seen in older adults) the definition most used in the studies is the co-occurrence of two or more chronic conditions. In the studies, the difference in prevalence between ≥2 and ≥3 multimorbidities in general is 12.9%.[15] Another proposed definition that aims to identify those individuals who need more complex healthcare is complex multimorbidity, defined as the 'co-occurrence of three or more chronic conditions that affect three or more different body systems in a person, without defining a chronic index condition'.[7] Thus, this review will include studies that defined multimorbidity as ≥2 and ≥3 chronic conditions, and complex multimorbidity, in order to ensure a better methodological agreement and increase the possibility of conducting a meta-analysis.

Studies with incomplete data, review articles, ecological studies, case reports or case series, randomised clinical trials and duplicate and unavailable data even after contacting the authors will be excluded. Studies addressing an index disease (such as cancer, heart disease or depression) or another age group will also be excluded, unless the groups were stratified so that data on older adults could be extracted and different outcomes could be analysed together (eg, hospitalisation and functional capacity).

## Training of the reviewers

The authors responsible for conducting the search and the inclusion/exclusion stage of the studies (LPR and ATdOR) will be appropriately trained. The application of the eligibility criteria will be tested in 50 titles and abstracts of the articles obtained from the search strategy. Subsequently, the reviewers will also receive training to identify the risk of bias in the studies included. For this step, Mendeley and Rayyan software programs will be used[37] to manage references and control the procedures performed.

## Review process

After searching the databases, duplicate studies will be deleted using Mendeley software. Two independent authors (LPR and ATdOR) will then read the titles and abstracts of all articles selected in the database search process using Rayyan software. Disagreements will be discussed and resolved by a third reviewer (EAS).

Agreement between reviewers will be measured using Cohen's kappa statistic,[38] in which values closer to 1 (>0.6) indicate a higher level of agreement between reviewers. The studies selected in the earlier stages (title and abstract screening) will also be read in full and evaluated according to the eligibility criteria by two independents authors (LPR and ATdOR). If any data that is important to the study is unavailable in the article, one of the researchers will directly contact the respective authors for clarification. After all of the above stages have taken

**Table 1** Form for extracting data on the impact of multimorbidity on hospitalisations in older adults

| Author Year Location | Type of study | Target population* | Multimorbidity† | Multimorbidity assessment method‡ | Hospitalisation† | Impact of multimorbidity on hospitalisations§ | Impact of multimorbidity on hospitalisations¶ | Type of analysis performed and adjustments for potential confounders |
|---|---|---|---|---|---|---|---|---|
| | | | | | | | | |
| | | | | | | | | |
| | | | | | | | | |

*Institutionalised or from the community.
†Definition, prevalence and CI of the prevalence.
‡Data source and multimorbidity measures used to investigate chronic disease (simple list of diseases, precompiled list of conditions or ad hoc tools).
§In-hospitalised subjects, by length of hospitalisation and readmission (measure of OR impact, prevalence ratio, HR—with the respective CI).
¶According to sex, age group and socioeconomic status.

place, those studies considered eligible will be included in the systematic review. A flow chart of the planned review process is shown in figure 1.

### Data extraction and evaluation of the study quality

Data will be extracted using a standardised form prepared by the authors, evaluating: author/year/location (city/country), type of study, target population (number of subjects, age group, sex, socioeconomic status/income of the country where the study was conducted or the income level of older adults, if analysed), the definition of multimorbidity used (≥2, ≥3, or complex multimorbidity; and the exact number and type of chronic conditions considered), multimorbidity assessment method (data source and multimorbidity measures used to assess chronic disease), co-occurrence of hospitalisation (yes/no), and the number and days of hospitalisations and readmissions. The impact of multimorbidity on hospitalisations, that is, length and number, will be measured by prevalence ratios, incidence ratios, ORs, HRs, relative risk (RR) and the respective 95% I and p values (table 1). If necessary, additional columns will be added to the data extraction table.

The Downs & Black Scale will be used[39] to analyse the risk of bias during study selection. However, not all items on the list are applicable, with only 19 of them relevant: 1–3, 5–7, 9–12, 15–18, 20, 21 and 25–27. Those with a score higher than 70% will be considered as having a low risk of bias. The Grading of Recommendations, Assessment, Development and Evaluation will be used to assess the quality of the evidence of the selected studies. The quality of evidence will be classified into four grades: high, moderate, low or very low.[40]

### Statistical analysis

The evidence on the impact of multimorbidity in the occurrence of hospitalisation in older adults will be summarised. In addition, a meta-analysis of the mean length of stay and the occurrence of hospital readmission will be carried out. Forest plot of random effects model will be built using ORs and their respective 95% CI, both for the meta-analysis of multimorbidity impact in the occurrence of hospitalisation and for analysing the occurrence of hospital readmission. RR, prevalence ratios and HRs and their respective 95% CI will be converted into ORs and included in the analysis. The combined results will be stratified according to socioeconomic status, institutionalisation, age and sex. The proposed statistical analysis will be performed using Stata V.14 SE (StataCorp).

The heterogeneity will be quantified using $I^2$[41] and a forest graphic will graphically display the effect sizes among studies.[42] For the studies with $I^2$ values greater than or equal 50%,[43] meta-regression may be performed according to the exposure variables described above. Publication bias will be assessed through a funnel plot[44] and asymmetry of the funnel plot will be evaluated using the Egger test.[45]

## DISCUSSION

Studies conducted in older adults are of major importance, as the impact on health and the risks in this age group are different from those in the general population. Understanding the impact of multimorbidity on hospitalisations later in life will provide guidance to healthcare professionals, as well as the development of public policies aimed at older adults with multimorbidity. The results of this study may also contribute to improving the management of national health agencies and private healthcare systems, thus contributing to a reduction of costs experienced by the healthcare system.

This systematic review study protocol describes the first study to investigate the impact of multimorbidity on hospitalisations in older adults as well as aspects related to this age group, particularly those aged 80 and older, differentiated in terms of sex and income of the population or income level of the country surveyed. This study will facilitate a deeper understanding of this topic in this age group and will support future research, reinforcing the importance of studies on this subject. Strengths of this study include the absence of restrictions in terms of language and year of publication, the use of methodological procedures that will assess the risk of bias in selecting studies, and inclusion of the three most commonly used definitions of multimorbidity. Potential limitations include the methodological diversity used in research involving multimorbidity, such as differences in the amount and type of chronic conditions included in the studies, and the stratification by age groups that may make it difficult to conduct a meta-analysis.

The results of this review may provide the basis for widening knowledge and to increase awareness of the main issues the association between multimorbidity and hospitalisation in older adults. Our findings may also contribute to improvements in public health policies resulting in cost reductions across healthcare systems. Ultimately, this review will seek to guide future research on the topic, identifying possible gaps and limitations of the existing literature.

**Correction notice** This article has been corrected since it was first published. The license type has been updated to CC BY.

**Contributors** EAS, LPR and ATdOR formulated to the conception and design of the study; EAS, LPR, ATdOR, MN and CdO contributed to the methodologic aspects; EAS, LPR, ATdOR, LNM, BPN, and MN wrote the protocol; EAS, CdO and MN carried out a final review of the protocol.

**Funding** This study was funded by The Economic and Social Research Council UK (grantES/T008822/1).

**Competing interests** None declared.

**Patient and public involvement** Patients and/or the public were not involved in the design, or conduct, or reporting, or dissemination plans of this research.

**Patient consent for publication** Not required.

**Provenance and peer review** Not commissioned; externally peer reviewed.

**Open access** This is an open access article distributed in accordance with the Creative Commons Attribution 4.0 Unported (CC BY 4.0) license, which permits others to copy, redistribute, remix, transform and build upon this work for any purpose, provided the original work is properly cited, a link to the licence is given, and indication of whether changes were made. See: https://creativecommons.org/licenses/by/4.0/.

**ORCID iDs**
Luciana Pereira Rodrigues http://orcid.org/0000-0002-6499-1451
Bruno Pereira Nunes http://orcid.org/0000-0002-4496-4122
Erika Aparecida Silveira http://orcid.org/0000-0002-8839-4520

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
