## [Reviewer comments · BMJ Open]

ARTICLE DETAILS

TITLE (PROVISIONAL)	What is the impact of multimorbidity on the risk of hospitalization in older adults? A systematic review study protocol
AUTHORS	Rodrigues, Luciana; Rezende, Andréa; Nogueira, Letícia; Nunes, BP; Noll, Matias; de Oliveira, Cesar; Silveira, Erika

VERSION 1 – REVIEW

REVIEWER	Maxwell, Colleen University of Waterloo, School of Pharmacy
REVIEW RETURNED	05-Mar-2021

GENERAL COMMENTS	Major Issues: 1) In the Abstract the authors mention the aim being to examine the impact of multimorbidity on the incidence of hospitalization in older adults; however they note in their Methods that they will include cross-sectional studies that would not allow for an understanding of temporal ordering of the observed association?2) Also in the Abstract (and elsewhere in the main paper), the authors note that the aim of the study is to improve the treatment of older adults with multimorbidity and to avoid hospitalization - but, how will the data they plan to summarize in their review actually contribute to such an aim? (e.g., they are not examining the association with potentially avoidable or preventable hospitalizations).3) In the Methods of the paper, it is noted that the searches will be carried out between Dec 2020 and Feb 2021; however, it is unclear what the time frame is for study inclusion?4) In the Methods of the paper, the authors note dual independent reviewers of Titles & Abstracts, but it is not clear that there will also be dual independent reviewers of eligible full papers and for data extraction, can the authors clarify these points?5) In the Statistical Analysis section, the authors have not described their planned stratified analyses. Nor have the authors commented on specific exploration of clinical heterogeneity. Minor Issues: 1) The authors may want to also capture measure of prolonged hospitalization or alternate level of care (ALC) stay (as any ALC bed stay or longer ALC bed stay is likely to be related to multimorbidity and also to patient gender).
--

REVIEWER	Zucchelli, Alberto
-----------------	--------------------

	University of Brescia
REVIEW RETURNED	18-Mar-2021

GENERAL COMMENTS	The Authors present the protocol for systematic review and meta-analysis aimed at exploring the relationship between multimorbidity and hospitalization. The resulting paper is likely to be important for actual literature about multimorbidity, given the high prevalence of this condition. The protocol is generally well written, although I suggest 1) to review the use of the word “concomitant” that is repeat several times and 2) to substitute “elderly” with “older persons” or “older patients” whenever possible. Comments:  • The Authors state that the aim of the study is to analyse the impact of multimorbidity on the incidence of hospitalization. However, they include cross-sectional studies in their protocol: because of the “snapshot” nature of this design, cross-sectional studies cannot contain such information. Due to the general idea of the study (multimorbidity as a risk factor for hospitalization), I suggest including only longitudinal studies in the present protocol. • Following the previous comment, if a study assesses the number of hospitalizations in the previous “year” for persons affected and not affected by multimorbidity, would it be included in the present study? A more detailed definition of the “impact on hospitalizations” and its relationship with the time frame of the study would help the reader to understand the study methodology. • I would include hazard ratios among the possible measures of association between multimorbidity and hospitalization. • It would be interesting to gain information about the way multimorbidity was assessed in each study. Chronic conditions can be investigated using self-administered questionnaires, registry data, comprehensive evaluation of medical history by nurses or physicians completed with the revision of medication. Furthermore, multimorbidity can be derived from a simple list of diseases, from a pre-compiled list of conditions a priori decided by the investigators or using ad hoc tools (such as the Charlson’s comorbidity index). The way multimorbidity is assessed has been proven to have an important impact on its prevalence. Information about multimorbidity assessment might help to direct future studies toward a more standardize way to investigate this important condition. Furthermore, this information might be used to stratify the analyses in case of conflicting results. • The Authors state that they will investigate whether the impact of multimorbidity on the incidence of hospitalization varies by gender, age, institutionalization, and socioeconomic status. Do the Authors plan to use such variables to run stratified analyses? • The Authors should clarify how they will report and handle measures of the association between multimorbidity and hospitalization adjusted for 1 or more confounders.
---

VERSION 1 – AUTHOR RESPONSE

Reports Reviewer 1: Prof. Colleen Maxwell, University of Waterloo, ICES

Major Issues:

1-In the Abstract the authors mention the aim being to examine the impact of multimorbidity on the incidence of hospitalization in older adults; however, they note in their Methods that they will include cross-sectional studies that would not allow for an understanding of temporal ordering of the observed association?

Response: Thank you for raising this important point. We agree with your consideration. For this reason, we have changed the aim to ‘the impact of multimorbidity on the occurrence of hospitalization in older adults’ (Lines 10 and 12, page 2 and lines 7 and 9, page 5).

2) Also in the Abstract (and elsewhere in the main paper), the authors note that the aim of the study is to improve the treatment of older adults with multimorbidity and to avoid hospitalization - but, how will the data they plan to summarize in their review actually contribute to such an aim? (e.g., they are not examining the association with potentially avoidable or preventable hospitalizations).

Response: Thank you for your comment. Since we will not examine the association with potentially avoidable or preventable hospitalizations, we have removed this aspect from our aim in the abstract as well as in the main paper. Moreover, we have included other relevant aspects, as follows: Lines 21 to 24, page 2: “The results of this review may provide the basis for widening knowledge and to increase awareness of the main issues the association between multimorbidity and hospitalization in older adults. Our findings may also contribute to improvements in public health policies resulting in cost reductions across healthcare systems. Ultimately, this review will seek to guide future research on the topic, identifying possible gaps and limitations of the existing literature.”

Lines 17 and 19, page 9: “Understanding the impact of multimorbidity on hospitalizations later in life will provide guidance to healthcare professionals, as well as the development of public policies aimed at older adults with multimorbidity”.

3) In the Methods of the paper, it is noted that the searches will be carried out between Dec 2020 and Feb 2021; however, it is unclear what the time frame is for study inclusion?

Response: Thank you. We have included this information as suggested and also updated the date, as follows:

Lines 24 to 27, page 5: “From February 2021 to April 2021, searches will be carried out in the PubMed, Embase, and Scopus databases by two independent researchers. There will be no restriction on the language and year of publication of the included studies and articles published until 30 April 2021 will be considered”.

4) In the Methods of the paper, the authors note dual independent reviewers of Titles & Abstracts, but it is not clear that there will also be dual independent reviewers of eligible full papers and for data extraction, can the authors clarify these points?

Response: Thank you. We have included this information in the revised version and clarified that dual independent reviewers will assess the full papers’ eligibility and data extraction, as follows:

Lines 27 to 29, page 7: “The studies selected in the earlier stages (title and abstract screening) will also be read in full and evaluated according to the eligibility criteria by two independent authors (LPR and ATOR)”.

5) In the Statistical Analysis section, the authors have not described their planned stratified analyses. Nor have the authors commented on specific exploration of clinical heterogeneity.

Response: Thank you for your relevant comment. We have improved the section about stratified meta-analysis and heterogeneity, as follows:

Lines 25 to 30, page 8 and lines 1 to 10, page 9: “The evidence on the impact of multimorbidity in the occurrence of hospitalization in older adults will be summarized. In addition, a meta-analysis of the

mean length of stay and the occurrence of hospital readmission will be carried out. Forest plot of random effects model will be built using odds ratios (OR) and their respective 95% confidence intervals (CI), both for the meta-analysis of multimorbidity impact in the occurrence of hospitalization and for analyzing the occurrence of hospital readmission. Relative risk (RR), prevalence ratios and hazard ratios and their respective 95% confidence intervals (CI) will be converted into ORs and included in the analysis. The combined results will be stratified according to socioeconomic status, institutionalization, age and sex. The proposed statistical analysis will be performed using Stata14® SE (Stata Corp, College Station, TX, USA).

The heterogeneity will be quantified using I² (41) and a forest graphic will graphically display the effect sizes among studies (42). For the studies with I² values greater than or equal 50% (43), meta-regression may be performed according to the exposure variables described above. Publication bias will be assessed through a funnel plot (44) and asymmetry of the funnel plot will be evaluated using the Egger test (45).”

Minor Issues:

6) The authors may want to also capture measure of prolonged hospitalization or alternate level of care (ALC) stay (as any ALC bed stay or longer ALC bed stay is likely to be related to multimorbidity and also to patient gender).

Response: Thank you for this relevant comment. Although this suggestion could be useful in further research, our main focus in this current protocol is to know whether the occurrence of multimorbidity increases the number of hospitalizations, length of stay and readmission.

Reviewer 2: Dr. Alberto Zucchelli, University of Brescia

Comments to the Author: The Authors present the protocol for systematic review and meta-analysis aimed at exploring the relationship between multimorbidity and hospitalization.

The resulting paper is likely to be important for actual literature about multimorbidity, given the high prevalence of this condition.

Response: Thank you for your positive feedback and kind words. Much appreciated.

The protocol is generally well written, although I suggest 1) to review the use of the word “concomitant” that is repeat several times and 2) to substitute “elderly” with “older persons” or “older patients” whenever possible.

Response: Thank you for this important observation. Regarding the use of the term ‘concomitant’, we have replaced it with other terms throughout the revised text. We have replaced ‘elderly’ by ‘older adults’.

3) The Authors state that the aim of the study is to analyse the impact of multimorbidity on the incidence of hospitalization. However, they include cross-sectional studies in their protocol: because of the “snapshot” nature of this design, cross-sectional studies cannot contain such information. Due to the general idea of the study (multimorbidity as a risk factor for hospitalization), I suggest including only longitudinal studies in the present protocol.

Response: Thank you for raising this important point. We did an exploratory literature review to write the protocol and we observed that the number of cohort studies is much lower than cross-sectional studies. Because of this, we decided to include both study designs to maximise our chances to cover relevant articles related to the theme. If we only include cohort studies our body of evidence would be limited. For this reason, we changed the aim to “the impact of multimorbidity on the occurrence of hospitalization in older adults”. We will also evaluate all the measures of association used in the published articles and include in the systematic review (such as: OR, RR, HR, etc) despite the temporality limitation of cross-sectional studies (Lines 10 and 12, page 2 and lines 7 and 9, page 5).

4) Following the previous comment, if a study assesses the number of hospitalizations in the previous “year” for persons affected and not affected by multimorbidity, would it be included in the present study? A more detailed definition of the “impact on hospitalizations” and its relationship with the time frame of the study would help the reader to understand the study methodology.

Response: Thank you for your comment. Taking into consideration your previous comment related to the inclusion of cross-sectional studies, we substituted the term ‘incidence’ with ‘occurrence’. Therefore, if a study assesses the number of hospitalizations in the previous ‘year’ for persons affected and not affected by multimorbidity, it will be included in the present review.

5) I would include hazard ratios among the possible measures of association between multimorbidity and hospitalization.

Response: Thank you. We have included hazard ratios among the possible measures of association between multimorbidity and hospitalization (Line 20, page 2 and line 14, page 8).

6) It would be interesting to gain information about the way multimorbidity was assessed in each study. Chronic conditions can be investigated using self-administered questionnaires, registry data, comprehensive evaluation of medical history by nurses or physicians completed with the revision of medication. Furthermore, multimorbidity can be derived from a simple list of diseases, from a pre-compiled list of conditions a priori decided by the investigators or using ad hoc tools (such as the Charlson’s comorbidity index). The way multimorbidity is assessed has been proven to have an important impact on its prevalence. Information about multimorbidity assessment might help to direct future studies toward a more standardized way to investigate this important condition. Furthermore, this information might be used to stratify the analyses in case of conflicting results.

Response: Thank you for raising this important point. We have included in the revised version a column entitled ‘multimorbidity assessment method’ in our Table 2 to consider the aspects raised by the reviewer, as follows:

Lines 19 to 28, page 14:

Table 2. Form for extracting data on the impact of multimorbidity on hospitalizations in older adults.

Author Year Location	Type of study	Target population ^a	Multimorbidity ^b	Multimorbidity assessment method ^c	Hospitalization ^b	Impact of multimorbidity on hospitalizations ^d	Impact of multimorbidity on hospitalizations ^e	Type of analysis performed and adjustments for potential confounders

^aInstitutionalized or from the community

^bDefinition, prevalence, and confidence interval of the prevalence

^cData source and multimorbidity measures used to investigate chronic disease (simple list of diseases, pre-compiled list of conditions or ad hoc tools).

^dIn hospitalized subjects, by length of hospitalization and readmission (measure of odds ratio impact, prevalence ratio, hazard ratio - with the respective confidence interval)

^eAccording to sex, age group, and socioeconomic status

7) The Authors state that they will investigate whether the impact of multimorbidity on the incidence of hospitalization varies by gender, age, institutionalization, and socioeconomic status. Do the Authors plan to use such variables to run stratified analyses?

Response: Thank you for the opportunity to clarify this point. We plan to include these specific variables in the impact of multimorbidity on the hospitalization in the extraction data Table and to run a meta-analysis for these subgroups.

8) The Authors should clarify how they will report and handle measures of the association between multimorbidity and hospitalization adjusted for 1 or more confounders.

Response: Thank you for your comment. We will report all the confounders that were used to adjust the association between multimorbidity and hospitalization. We will also report the type of statistical analyses performed in each study in a column entitled "Type of analysis performed and adjustments for potential confounders". We intend to verify which confounders are more commonly used to adjust the associations. We will investigate whether the studies have used confounders to evaluate the risk of bias and the methodological quality of the articles. We have expanded the statistical analysis section as follows:

Lines 25 to 30, page 8 and lines 1 to 10, page 9: "The evidence about multimorbidity impact in the occurrence of hospitalization in older adults will be summarized as well as a meta-analysis of the mean length of stay and the occurrence of hospital readmission will be carried on. A forest plot of random effects model will be built using odds ratios (OR) and their respective 95% confidence intervals (CI), both for the meta-analysis of multimorbidity impact in the occurrence of hospitalization and for analysing the occurrence of hospital readmission. Relative risk (RR), prevalence ratios and hazard ratios and their respective 95% confidence intervals (CI) will be converted into ORs and included in the analysis. The combined results will be stratified according to socioeconomic status, institutionalization, age and sex. The proposed statistical analysis will be performed using Stata14® SE (Stata Corp, College Station, TX, USA).

The heterogeneity will be quantified using I^2 (41) and a forest graphic will graphically display the effect sizes among studies (42). For the studies with I^2 values greater than or equal 50% (43), meta-regression may be performed according to the exposure variables described above. Publication bias will be assessed through a funnel plot (44) and asymmetry of the funnel plot will be evaluated using the Egger test (45)."